# Filamin A Is a Prognostic Serum Biomarker for Differentiating Benign Prostatic Hyperplasia from Prostate Cancer in Caucasian and African American Men

**DOI:** 10.3390/cancers16040712

**Published:** 2024-02-08

**Authors:** Nischal Mahaveer Chand, Poornima K. Tekumalla, Matt T. Rosenberg, Albert Dobi, Amina Ali, Gregory M. Miller, Juan J. Aristizabal-Henao, Elder Granger, Stephen J. Freedland, Mark D. Kellogg, Shiv Srivastava, David G. McLeod, Niven R. Narain, Michael A. Kiebish

**Affiliations:** 1BPGbio Inc., Framingham, MA 01701, USA; nischal.chand@bpgbio.com (N.M.C.); poornima.tekumalla@bpgbio.com (P.K.T.); greg.miller@bpgbio.com (G.M.M.); juan.henao@bpgbio.com (J.J.A.-H.); elder.granger@gmail.com (E.G.); niven.narain@bpgbio.com (N.R.N.); 2Mid Michigan Health Centers, Jackson, MI 49201, USA; matttoren@yahoo.com; 3Center for Prostate Disease Research, John P. Murtha Cancer Center Research Program, Department of Surgery, Uniformed Services University of the Health Sciences and the Walter Reed National Military Medical Center, Bethesda, MD 20817, USA; adobi@cpdr.org (A.D.); amina.m.ali2.ctr@mail.mil (A.A.);; 4Henry M. Jackson Foundation for the Advancement of Military Medicine, Bethesda, MD 20817, USA; 5Center for Integrated Research in Cancer and Lifestyle, Cedars-Sinai Medical Center, Los Angeles, CA 90048, USA; stephen.freedland@cshs.org; 6Department of Laboratory Medicine, Boston Children’s Hospital Department of Pathology, Harvard Medical School, Boston, MA 02115, USA; mark.kellogg@childrens.harvard.edu; 7Department of Biochemistry and Molecular & Cell Biology, Georgetown University School of Medicine, Washington, DC 20057, USA; shsr629@gmail.com

**Keywords:** prostate cancer, benign prostatic hyperplasia, prostate specific antigen, biomarker, FLNA

## Abstract

**Simple Summary:**

Prostate Cancer represents a significant health risk for men, especially African American men, despite the availability of PSA testing. Although PSA testing is the current gold-standard test for identifying at-risk men, an increased PSA level may arise from Benign Prostatic Hyperplasia instead of Prostate Cancer. For men with BPH, PSA testing may lead them to undergo unnecessary biopsies. As an alternative to PSA, we have previously described a Filamin-A and prostate volume based biomarker test with superior performance. To simplify this test, we removed the requirement of prostate volume measurement. Herein, we present results of this updated test utilizing Filamin-A alone in Caucasian and African American men. Filamin-A demonstrates superior predictive power compared to PSA in both patient populations. By reliably separating benign conditions from aggressive prostate cancer, this test would reduce the health care burden resulting from unnecessary prostate biopsies.

**Abstract:**

Prostate cancer represents a significant health risk to aging men, in which diagnostic challenges to the identification of aggressive cancers remain unmet. Prostate cancer screening is driven by the prostate-specific antigen (PSA); however, in men with benign prostatic hyperplasia (BPH) due to an enlarged prostate and elevated PSA, PSA’s screening utility is diminished, resulting in many unnecessary biopsies. To address this issue, we previously identified a cleaved fragment of Filamin A (FLNA) protein (as measured with IP-MRM mass spectrometry assessment as a prognostic biomarker for stratifying BPH from prostate cancer and subsequently evaluated its expanded utility in Caucasian (CA) and African American (AA) men. All men had a negative digital rectal examination (DRE) and PSA between 4 and 10 ng/mL and underwent prostate biopsy. In AA men, FLNA serum levels exhibited diagnostic utility for stratifying BPH from patients with aggressive prostate cancer (0.71 AUC and 12.2 OR in 48 men with BPH and 60 men with PCa) and outperformed PSA (0.50 AUC, 2.2 OR). In CA men, FLNA serum levels also exhibited diagnostic utility for stratifying BPH from patients with aggressive prostate cancer (0.74 AUC and 19.4 OR in 191 men with BPH and 109 men with PCa) and outperformed PSA (0.46 AUC, 0.32 OR). Herein, we established FLNA alone as a serum biomarker for stratifying men with BPH vs. those with high Gleason (7–10) prostate cancers compared to the current diagnostic paradigm of using PSA. This approach demonstrates clinical actionability of FLNA alone without the requirement of prostate volume measurement as a test with utility in AA and CA men and represents a significant opportunity to decrease the number of unnecessary biopsies in aggressive prostate cancer diagnoses.

## 1. Introduction

The detection and diagnosis of aggressive prostate cancer (PCa) remains a critical medical need due to its incidence and mortality in the global male population. According to recent projections in the United States population, PCa is the most common cancer diagnosis in men, accounting for 29% of all new cancer cases, and is the second leading cause of mortality in men following lung cancer [1]. For this reason, effective screening tools are needed to detect life threatening PCa accurately, while simultaneously avoiding unnecessary biopsies in low-risk populations. The prostate-specific antigen (PSA) is one such screening tool widely used in clinical practice. Unfortunately, the adoption of PSA testing has led to unnecessary treatment of men with low-risk PCa due to its low sensitivity and specificity in detecting clinically significant PCa in men with elevated PSA levels. The lack of specificity of PSA is largely driven by the incidence of benign prostatic hyperplasia (BPH), a non-cancerous condition also characterized by elevated PSA levels [2,3]. Therefore, there is a critical unmet need for improved methods to distinguish patients with BPH from patients with clinically significant PCa.

We previously identified the level of a cleaved peptide of Filamin A (FLNA) in serum, when combined with prostate volume and age among men who all had elevated PSA levels between 4 and 10 ng/mL, that provided superior predictive performance compared to PSA in distinguishing between men with BPH and men with PCa [4]. The identified protein, FLNA, has been implicated for its role in PCa metastasis, which supports the biological plausibility that FLNA may be a serum biomarker for a PCa diagnosis [5].

Further, we previously reported that the inclusion of prostate volume as an additional variable with FLNA provided superior predictive power for diagnostics. Prostate volume is quantified via transrectal ultrasound (TRUS); however, TRUS can pose some challenges regarding obtaining accurate volumetric measurements due to human error and is found to be highly user-dependent [6,7]. Additionally, TRUS exposes patients to unnecessary procedures that might result in complications [8,9]. A diagnostic test that eliminates the need for TRUS would be advantageous for the patient and be more cost-effective. For this reason, we report results from a reanalysis of our previous cohort without prostate volume and establish FLNA alone to be a reliable test while still providing better predictive performance compared to PSA alone in two different ethnic populations.

Finally, the development of diagnostics has historically focused primarily on Caucasian (CA) populations, while largely lacking in studies of populations of different ancestral origins, especially underserved populations such as Hispanic Americans, Native Americans, and African Americans (AAs). Recent work has illuminated the disproportionate impact of PCa on AA men: (1) AA men are more likely to harbor genomically aggressive cancer [10]; (2) AA men have a two to four times higher PCa mortality rate than other racial and ethnic groups [1]; (3) annual screening for PCa would be especially beneficial in AA men compared to CA men [11]. Given the impact of PCa on underserved populations, there is an unmet need for a novel PCa screening or reflex tool capable of identifying at-risk patients across men of different ancestry. Toward this aim, we evaluated the utility of FLNA alone without other risk factors in both CA men and AA men separately and found that FLNA provides superior performance in distinguishing men with BPH from men with PCa as compared to PSA in both patient groups.

## 2. Methods

This study analyzed a previously published cohort of men whose age ranged between 35 and 92 years old and who had a median age of 63 years and analyzed a specific combination of markers for the investigation of its utility in Caucasian as well as African American men. Within this study cohort, 300 men had BPH (191 CA, 48 AA, and 61 other-race), while 477 men had PCa (281 CA, 139 AA, and 57 other-race) (Table 1). Men with BPH were subjected to at least one biopsy to indicate their PCa-negative status. In contrast, men with PCa were confirmed with biopsy. All men in the analysis cohort had a PSA in the range of 4–10 ng/mL and a negative digital rectal exam (DRE). Additional clinical characteristics of the study cohort are described in Table 1, also being previously published in [4].

In our previous study, we showed that logistic regression using a panel of FLNA, age, and prostate volume performed better than PSA in discriminating between men with PCa and men with BPH. Here, we evaluated five parsimonious logistic regression models comprised of at most two of these features (FLNA alone, PSA alone, age alone, FLNA and age, and PSA and age) and compared their performance to our previous model using FLNA, age, and prostate volume. Models were evaluated by their capability to distinguish PCa (aggressive, i.e., Gleason ≥ 7, as well as comparison of all patients with PCa) from BPH using AUC as a performance metric computed at each model’s optimal cutoff, determined at a sensitivity ≥ 0.9. Finally, model performances were assessed on CA and AA patient subsets separately for each classification analysis.

### Quantitation of FLNA

**Antibody immobilization.** Three mouse monoclonal antibodies, Anti-FLNA 2C12 and Anti-FLNA 3F4, were immobilized using the Thermo Fisher Scientific Pierce Direct IP Kit (Thermo Fisher Scientific, Waltham, MA, USA) according to the manufacturer’s recommendations, with a few modifications as previously described [12]. In total, 200 µg of each of the antibodies was coupled individually to 200 µL of AminoLink Plus coupling resin and stored at 4 °C until needed.

**Immunoprecipitation calibration standard generation.** Immunoprecipitation tubes were prepared by aliquoting 5 µL of each of the two antibody-coupled resins into the IP tube (Pierce Direct IP Kit, Thermo Fisher Scientific). The resin was washed twice with 200 µL of an IP lysis/wash buffer. In total, 100 µL of a human serum sample or 100 µL of water (surrogate matrix) was added to each IP tube along with 500 µL of a prepared lysis buffer solution (IP lysis/wash buffer with 1.2X Halt protease cocktail inhibitor; Thermo Fisher Scientific) and 0.5 M EDTA, then incubated overnight at 4 °C with end-over-end mixing. The resin was washed five times with 200 µL of the IP lysis/wash buffer and once with 100 µL of a 1X conditioning buffer. The captured proteins were eluted with 50 µL of an elution buffer with an incubation time of 15 min, and then neutralized with 5 µL of 1 M Tris HCl, pH 9.0 (Teknova, Hollister, CA, USA). The IP eluates from the surrogate matrix were used to prepare P2 (AGVAPLQV) peptide calibration curves by spiking with a P2 synthetic peptide (Genscript, Piscataway, NJ, USA) stock solution (0.2/0.36 µg/mL) followed by serial dilution. P2 calibration standards ranged from 125 pg/mL to 2000 pg/mL. All samples were then subjected to trypsin digestion as described below.

**Trypsin Digestion of IP-extracted samples.** Trypsin digestion was performed using the Flash Digest Kit (Perfinity Biosciences, West Lafayette, IN, USA) following the manufacturer’s protocol with few modifications. Flash digest tubes were equilibrated to room temperature, and then centrifuged for 1 min at 1500× *g* and 5 °C. In total, 50 µL of each sample, 25 µL of a digestion buffer (Perfinity Biosciences), and 5 µL of a working internal standard (Thermo Fisher Scientific) solution (P2/P4 10/30 ng/mL) were added to the Flash digest tubes. After vortexing, samples were digested at 70 °C for 20 min in Eppendorf Thermo Mixer C (Eppendorf, Framingham, MA USA). The Flash digest tubes were then centrifuged for 5 min at 1500× *g* and 5 °C. A 60 µL aliquot of the supernatant was transferred to an LC-MS vial.

**LC-MS/MS analysis.** MRM analyses were performed on a 6500 QTRAP mass spectrometer (SCIEX) equipped with an electrospray source, a 1290 Infinity UPLC system (Agilent Technologies, Santa Clara, CA, USA), and an Xbridge Peptide BEH300 C18 (3.5 μm, 2.1 mm × 150 mm) column (Waters, Milford, MA, USA). Liquid chromatography was carried out at a flow rate of 400 µL/min, and the sample injection volume was 30 µL. The column was maintained at a temperature of 60 °C. Mobile phase A consisted of 0.1% formic acid (Sigma Aldrich, St. Louis, MO, USA) in water (Thermo Fisher Scientific), and mobile phase B consisted of 0.1% formic acid in acetonitrile (Thermo Fisher Scientific). The gradient with respect to %B was as follows: 0–1.5 min, 5%; 1.5–2 min, 5–15%; 2–5 min, 15%; 5–7.1 min, 15–20%; 7.1–8.1 min, 20–80%; 8.1–9.0 min, 80%; 9.0–9.1 min, 80–5%; and 9.1–16 min, 5%. The instrument parameters for the 6500 QTRAP mass spectrometer were as follows: ion spray voltage of 5500 V, curtain gas of 20 psi, collision gas set to “medium”, interface heater temperature of 400 °C, nebulizer gas (GS1) of 80 psi and ion source gas (GS2) of 80 psi, and unit resolution for both Q1 and Q3 quadrupoles.

**PMRM peptide quantitation.** A data analysis was performed using Analyst^®^ software (version 1.6.2, SCIEX, Framingham, MA, USA) and peak integrations were reviewed manually. The calibration curve for the FLNA P2 peptide was constructed by plotting the peak area ratios (analyte/internal standard) versus concentration of the standard with 1/×2 linear least square regression. The regression equations from P2 calibration standards were used to back-calculate the measured P2 concentrations for each QC and unknown sample.

## 3. Results

### 3.1. Comparison of FLNA with PSA to Distinguish PCa from BPH Using Entire Study Cohort

We evaluated the ability of FLNA alone and PSA alone in distinguishing 477 men with PCa from 300 men with BPH along with the utility of using different variables to assess relationships involving PSA (Figure 1). This analysis presented FLNA as an independent variable from PSA as well as closely associated with the Gleason score in men with PCa, whereas PSA and prostate volume were associated in men with BPH. The FLNA biomarker yielded sensitivity greater than 0.9 with a positive predictive value (PPV) of 0.69 and a negative predictive value (NPV) of 0.72 in patients with prostate cancer with a range of Gleason scores, including low and high Gleason PCa (Table 2(A)). In contrast, PSA yielded a PPV of 0.6 and an NPV of 0.27. The diagnostic odds ratio (OR) of the FLNA test was 5.76 (CI: 3.78–8.94) with the *p*-value = 5.7 × 10^−19^. Additionally, the FLNA test provided an AUC of 0.65 as compared to an AUC of 0.55 for PSA.

Next, we compared the performance of FLNA with PSA on a different classification task: discriminating 477 men with PCa from 95 men with BPH who had more than one negative biopsy (analyzing the serum sample prior to the first biopsy). The rationale for this is that PCa can be missed on the first biopsy. Thus, for the 95 men with two or more negative biopsies, we can demonstrate improved insight that they do not harbor missed cancer, demonstrating that an independent biomarker other than PSA can provide diagnostic insight. Here, the FLNA diagnostics again provided superior predictive performance over PSA (Table 2(B)). The FLNA biomarker yielded sensitivity greater than 0.9 with a PPV of 0.92, an NPV of 0.61, an OR of 18.8 (CI: 10.8–33.3), a *p*-value = 1 × 10^−29^, and an AUC of 0.83. In comparison, PSA gave a PPV of 0.84, an NPV of 0.24, an OR of 1.71 (CI: 0.848–3.3), and an AUC of 0.57.

Finally, we considered a different classification task to compare FLNA with PSA: discriminating 203 men with PCa and higher-grade PCa (Gleason score between 7 and 10) from 300 men with BPH. FLNA provided superior predictive performance over PSA (Table 2(C)). The FLNA biomarker yielded sensitivity greater than 0.9 with a PPV of 0.49, an NPV of 0.91, an AUC of 0.68, an OR of 9.61 (CI: 4.83–21.3), and a *p*-value = 6.8 × 10^−16^. This is an improvement over PSA, which showed a PPV of 0.4, an NPV of 0.55, an OR of 0.796 (CI: 0.408–1.57), an AUC of 0.5, and a *p*-value of 0.52.

### 3.2. Comparison of FLNA with PSA in AA Men

We further assessed our comparison of FLNA and PSA in a patient subset of men of African descent (Table 3(A)). The FLNA biomarker yielded sensitivity greater than 0.9 with a PPV of 0.82, an NPV of 0.76, and an AUC of 0.72 when classifying 139 AA men with PCa (range of Gleason scores) from 48 AA men with BPH. Additionally, the OR of the FLNA biomarker was 14.2 (CI: 4.94– 47.5) with a *p*-value = 1.5 × 10^−8^. This performance was superior to PSA, which provided a PPV of 0.75, an NPV of 0.28, an AUC of 0.54, and a *p*-value of 0.78.

Additionally, with a focus on the AA patient subset, we evaluated the performance of FLNA as compared to PSA in classifying 139 AA men with PCa from 25 men with BPH who had more than one negative biopsy (Table 3(B)). The FLNA test showed sensitivity greater than 0.9 with a PPV of 0.93, an NPV of 0.71, and an AUC of 0.81 when classifying 139 AA men with PCa from 25 men with BPH and multiple biopsies. Furthermore, FLNA resulted in an OR of 31.7 (CI: 9.33–125) with a *p*-value = 1.8 × 10^−10^. In contrast, PSA performed worse with a PPV of 0.86, an NPV of 0.28, an AUC of 0.6, and a *p*-value of 0.16.

The further analysis compared the performance of FLNA with PSA in classifying 60 AA men with PCa and a Gleason score between 7 and 10 from 48 AA men with BPH (Table 3(C)). The FLNA biomarker performed well with a sensitivity greater than 0.9, a PPV of 0.66, an NPV of 0.86, and an AUC of 0.71 when classifying AA men with PCa and a Gleason score of 7–10 from AA men with BPH. In addition, the FLNA test demonstrated an OR of 12.2 (CI: 3.2–69.3) and a *p*-value = 1.7 × 10^−5^. FLNA performed favorably when compared to using PSA for this classification task, which yielded a PPV of 0.58, an NPV of 0.62, an AUC of 0.5, and a *p*-value of 0.24, which demonstrated minimal diagnostic utility in AA men.

### 3.3. Comparison of FLNA with PSA in CA Men Patient Subset

We performed an analysis to compare the performance of FLNA to PSA in a patient subset of CA men. The FLNA biomarker yielded sensitivity greater than 0.9, a PPV of 0.68, an NPV of 0.73, and an AUC of 0.68 in classifying 281 CA men with PCa from 191 CA men with BPH (Table 4(A)). Additionally, the FLNA biomarker had an OR of 5.64 (CI: 3.33–9.79), and a *p*-value = 1.6 × 10^−12^. In comparison, PSA performed worse on this classification task with a PPV of 0.58, an NPV of 0.28, an AUC of 0.56, and a *p*-value of 0.13.

Next, we evaluated the ability of FLNA as compared to PSA in the CA patient subset to classify 281 CA men with PCa from 67 CA men with BPH who had more than one negative biopsy (Table 4(B)). FLNA yielded sensitivity greater than 0.9, a PPV of 0.91, an NPV of 0.63, and an AUC of 0.84 in classifying CA men with PCa from CA men with BPH and multiple negative biopsies. Also, FLNA exhibited an OR of 18.1 (9.15–36.9) and a *p*-value = 1.5 × 10^−20^. PSA performed worse on this classification task with a PPV of 0.81, an NPV of 0.24, and an AUC of 0.59 with a *p*-value of 0.39.

Lastly, we compared FLNA to PSA in classifying 109 CA men with PCa and a Gleason score of 7–10 from 191 CA men with BPH (Table 4(C)). FLNA performed well, with a sensitivity greater than 0.9, a PPV of 0.46, an NPV of 0.96, and an AUC of 0.74. Additional characteristics of the FLNA model in this classification were an OR of 19.4 (CI: 6.08–99.3) and a *p*-value = 1.6 × 10^−12^. In contrast, PSA showed a PPV of 0.35, an NPV of 0.38, and an AUC of 0.46, an OR of 0.32, and a *p*-value of 0.033.

## 4. Discussion

PCa, due to its prevalence and mortality, has attracted significant interest for the development of novel diagnostic tools for the detection and identification of patients with aggressive cancers. Nonetheless, PSA testing remains the standard of care for PCa detection. In the United States, for instance, men over 55 years of age are recommended to discuss with their physician and undergo shared decision making regarding routine PSA testing despite the absence of symptoms for PCa [13]. Recent efforts have led to the development of PSA-derived screening tests such as the Prostate Health Index (PHI) as a secondary screening test [14]. However, the accuracy of using PSA to detect PCa is diminished in men with an enlarged prostate volume, leading to elevated PSA levels unrelated to PCa. This relationship involving elevated PSA arising from an increased prostate volume has been previously established [15,16]. Thus, the use of PSA, or PSA-derived tests, to detect PCa may lead to false positives, leading to unnecessary biopsies that create a significant burden on patient wellbeing and the healthcare system. A complementary test for PSA that overcomes its limitations, especially in the BPH population, and provides higher sensitivity is needed to improve patient wellbeing and reduce healthcare costs. Underserved populations such as AA men have an increased risk of PCa with AA men more likely to die at a younger age as compared to other ethnicities [1]. Due to underrepresentation in PCa diagnostic studies, whether these underserved populations are at a higher risk of either false positives or an increased likelihood of morality resulting from failure to detect severe disease false negatives using currently available biomarkers with a focus on separating BPH from men with PCa is unknown.

Emerging technologies for measuring circulating proteins and cleaved fragments of proteins, which may serve as biomarkers for diseases (e.g., cancer, cardiovascular disease, neurological disorders), are helping through advances in the field of proteomics by providing high-sensitivity and high-specificity biomarkers at increased throughput [17]. Notable examples of these emerging technologies are (1) Nanostring, a digital gene expression platform that measures mRNA levels, (2) IP-MRM, a mass-spectrometry-based method for quantifying low-abundance proteins, and (3) aptamer-based techniques that use single-stranded RNA or DNA molecules to bind specifically to target proteins [17]. In particular, the ability to measure cleaved protein fragments (e.g., FLNA) with high sensitivity has been improved with the use of IP-MRM [12]. The quantification of FLNA with IP-MRM provides a non-invasive method to determine the risk of aggressive PCa using serum samples, as opposed to more expensive methods such as MRI that are used to guide the biopsy, which would further decrease the barrier for their adoption regarding underserved and minority populations [18,19]. Further advantages of blood-based PCa diagnostics, such as FLNA, include the ability to use the diagnostics at multiple timepoints in the patient’s journey. For instance, the blood-based FLNA test may be used in the primary care setting as a secondary screening tool following PSA testing as a means for reducing the number of unnecessary biopsies for men with BPH. Additionally, urologists could utilize the blood-based FLNA test to quantify the risk of aggressive PCa and necessity for performing a biopsy.

The clinical unmet need to avoid unnecessary biopsies is significant. Biopsies can lead to various complications, such as bleeding and infection. Additionally, biopsies are invasive procedures that can cause pain, anxiety, and discomfort to patients. In the case of men with BPH, the presence of elevated PSA levels may lead to repeated unnecessary biopsies with consistent negative results. This not only exposes these men to the risks and complications associated with biopsy but also contributes to increased healthcare costs and the overutilization of medical resources. Therefore, avoiding unnecessary biopsies is crucial for improving patient outcomes and reducing healthcare costs. By developing a non-invasive method that is independent of PSA, such as the FLNA diagnostics described here, the rate of unnecessary biopsies may be reduced and the risk of aggressive PCa detection can be determined to allow healthcare providers to make an informed decision to pursue biopsies. Some limitations of the current FLNA diagnostics potentially include the widespread adoption of IP-MRM techniques in risk assessment for cancer diagnoses; however, this technique has been widely adopted for other diagnostic testing. Further, additional ethnicities such as Hispanic or Asian populations were not presently evaluated in the current study. While most diagnostic tests are based on PSA-derived factors, this potentially limits their use in distinguishing BPH from aggressive PCa as demonstrated with the current analysis. Additionally, there has been an ethnic prevalence on the markers in some tests, such as exosomeDX, in which a portion of the markers are more amplified in CA than AA men, which might limit their performance in some ethnic groups. All in all, there is a continued need to develop novel diagnostic tools to improve decision support to assess the risk for aggressive prostate cancer in diverse ethnic groups, and the assessment of FLNA may provide an opportunity to address this clinical unmet need.

## 5. Conclusions

In this work, we evaluated the performance of a revised PCa biomarker panel in a reanalysis of our previous study cohort. By removing prostate volume, which can be prone to variability, we evaluated the use of FLNA alone to demonstrate superior predictive performance compared to PSA alone in discriminating aggressive PCa from BPH. Furthermore, we found that FLNA was independent of PSA in PCa men, and FLNA was uncorrelated with Age and PV in BPH men.

Another important aspect of PCa diagnosis is the ethnic disparity in the incidence and mortality of the disease. AA men have the highest rate of PCa in the world and are more likely to develop aggressive PCa than CA men. Therefore, there is an urgent need to identify biomarkers that can accurately detect aggressive PCa in AA men as well as CA men, which would reduce the health disparity. In this study, we confirmed that FLNA alone had a high sensitivity and specificity in discriminating aggressive PCa from BPH in both AA and CA men. This suggests that FLNA is a robust and reliable biomarker that can be used across different ethnic populations to identify men who are at risk of having aggressive PCa. Unnecessary biopsies are a major concern in PCa diagnosis, as they can cause physical and psychological harms to patients, increase health care costs, and contribute to the overdiagnosis and overtreatment of indolent PCa. Therefore, developing a reliable and non-invasive test that can accurately detect aggressive PCa is of great clinical significance.

## Figures and Tables

**Figure 1 cancers-16-00712-f001:**
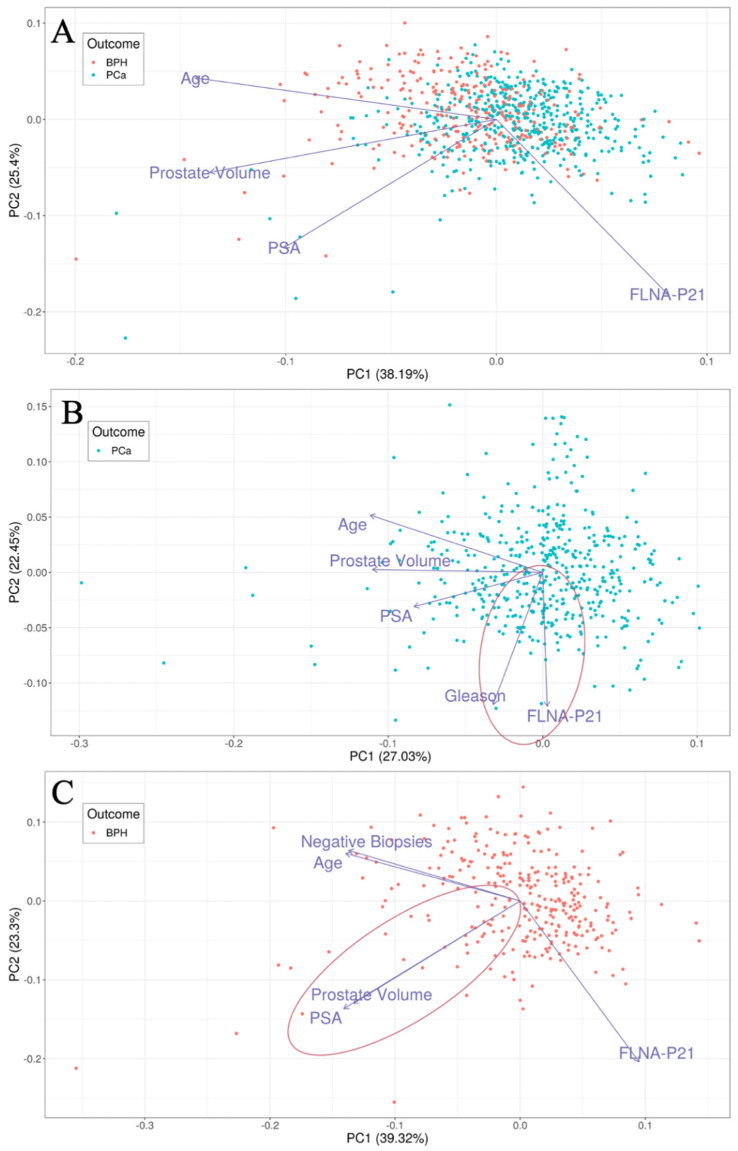
Principal component analysis plots showing first two principal components (PCs) for (**A**) patients with PCa and BPH (*n* = 777), (**B**) patients with PCa (*n* = 477), (**C**) patients with BPH (*n* = 300). In each plot, loadings of feature used for PCA are represented as vectors. Angles between any two loadings signify the correlation between the features (smaller angles = positive correlation; right angles = no correlation; wide angles = negative correlation). Highlighted in red are two correlations of importance; FLNA is positively correlated with Gleason in patients with PCa (panel **B**) and PSA is strongly positively correlated with prostate volume in patients with BPH (panel **C**).

**Table 1 cancers-16-00712-t001:** Demographic table for 777 patients. Gleason Scores are only available for patients with PCa and are not applicable for patients with BPH. Negative Biopsies are only available for patients with BPH; all patients with PCa underwent a positive biopsy.

Characteristic	BPH (*n* = 300)	PCa (*n* = 477)
Age (years) ^†^	65 (8)	62 (8)
Prostate Volume (mL) ^†^	61 (27)	42 (26)
PSA (ng/mL) ^†^	6.7 (4.7)	6.5 (5.9)
FLNA (ng/mL) ^†^	5.07 (0.55)	5.39 (0.51)
Race ^‡^		
African American	48 (16%)	139 (29%)
Caucasian	191 (64%)	281 (59%)
Other	61 (20%)	57 (12%)
Gleason Scores ^‡^		
Low (3–6)		238 (54%)
High (7–10)		203 (46%)
Negative Biopsies ^‡^		
Single	205 (68%)	
Multiple	95 (32%)	
Collection Site ^‡^		
Cleveland Clinic	5 (1.7%)	126 (26%)
CPDR/Walter Reed	119 (40%)	239 (50%)
Uni. Health Network	118 (39%)	78 (16%)
Veteran Affairs	58 (19%)	34 (7.1%)

^†^ Mean (std. dev) ^‡^ Count (%).

**Table 2 cancers-16-00712-t002:** Results of logistic models to distinguish patients with PCa from patients with BPH for (A) 477 patients with PCa from 300 patients with BPH, (B) 477 patients with PCa from 95 patients with BPH with multiple negative biopsies, (C) 203 patients with Gleason Scores 7–10 PCa from 300 patients with BPH.

(A) All Patients Comparing BPH vs. PCa										
Model	Cutoff	AUC	Sensitivity	Specificity	PPV	NPV	OR (CI)	*p*-value	nBPH	nPCa
PSA	0.61	0.55	0.9	0.057	0.6	0.27	0.55 (0.29–0.999)	0.044	300	477
Age	0.52	0.59	0.92	0.19	0.64	0.6	2.71 (1.71–4.33)	8.8 × 10^−6^	300	477
FLNA	0.5	0.65	0.92	0.33	0.69	0.72	5.76 (3.78–8.94)	5.7 × 10^−19^	300	477
PSA + Age	0.53	0.59	0.9	0.21	0.64	0.57	2.38 (1.55–3.68)	3.2 × 10^−5^	300	477
FLNA + Age	0.49	0.66	0.9	0.32	0.68	0.67	4.36 (2.92–6.58)	1.4 × 10^−14^	300	477
FLNA + Age + Vol	0.5	0.75	0.9	0.45	0.72	0.74	7.36 (4.99–11)	1.4 × 10^−28^	300	477
**(B) All Patients Undergoing Multiple Biopsies**										
**Model**	**Cutoff**	**AUC**	**Sensitivity**	**Specificity**	**PPV**	**NPV**	**OR (CI)**	***p*-value**	**nBPH**	**nPCa**
PSA	0.81	0.57	0.9	0.16	0.84	0.24	1.71 (0.848–3.3)	0.1	95	477
Age	0.72	0.69	0.92	0.31	0.87	0.43	5.06 (2.81–9.07)	2.5 × 10^−8^	95	477
FLNA	0.72	0.83	0.92	0.62	0.92	0.61	18.8 (10.8–33.3)	1 × 10^−29^	95	477
PSA + Age	0.73	0.7	0.9	0.31	0.87	0.38	4.01 (2.26–7.03)	1.1 × 10^−6^	95	477
FLNA + Age	0.7	0.85	0.9	0.62	0.92	0.56	14.9 (8.7–25.8)	1.2 × 10^−26^	95	477
FLNA + Age + Vol	0.71	0.87	0.9	0.67	0.93	0.58	18.7 (10.8–33)	4.2 × 10^−31^	95	477
**(C) All Patients Comparing BPH with PCa Gleason Score 7–10**										
**Model**	**Cutoff**	**AUC**	**Sensitivity**	**Specificity**	**PPV**	**NPV**	**OR (CI)**	***p*-value**	**nBPH**	**nPCa**
PSA	0.38	0.5	0.9	0.08	0.4	0.55	0.796 (0.408–1.57)	0.52	300	203
Age	0.35	0.55	0.92	0.15	0.42	0.73	1.98 (1.07–3.8)	0.027	300	203
FLNA	0.28	0.68	0.95	0.33	0.49	0.91	9.61 (4.83–21.3)	6.8 × 10^−16^	300	203
PSA + Age	0.34	0.55	0.9	0.16	0.42	0.71	1.74 (0.975–3.21)	0.062	300	203
FLNA + Age	0.28	0.68	0.9	0.33	0.48	0.83	4.56 (2.67–8.12)	3.6 × 10^−10^	300	203
FLNA + Age + Vol	0.29	0.77	0.9	0.44	0.52	0.87	7.07 (4.17–12.5)	3.4 × 10^−17^	300	203

**Table 3 cancers-16-00712-t003:** Results of logistic model to distinguish patients with PCa from patients with BPH among AA patients for (A) 139 patients with PCa from 48 patients with BPH, (B) 139 patients with PCa from 25 patients with BPH with multiple negative biopsies, (C) 60 patients with Gleason Scores 7–10 PCa from 48 patients with BPH.

(A) African American Patients Comparing BPH vs. PCa										
Model	Cutoff	AUC	Sensitivity	Specificity	PPV	NPV	OR (CI)	*p*-value	nBPH	nPCa
PSA	0.74	0.54	0.91	0.1	0.75	0.28	1.13 (0.297–3.62)	0.78	48	139
Age	0.58	0.74	0.91	0.4	0.81	0.59	6.27 (2.61–15.6)	7.7 × 10^−6^	48	139
FLNA	0.61	0.72	0.96	0.4	0.82	0.76	14.2 (4.94–47.5)	1.5 × 10^−8^	48	139
PSA + Age	0.56	0.74	0.91	0.4	0.81	0.59	6.27 (2.61–15.6)	7.7 × 10^−6^	48	139
FLNA + Age	0.57	0.79	0.91	0.5	0.84	0.65	9.53 (4.04–23.5)	1.3 × 10^−8^	48	139
FLNA + Age + Vol	0.6	0.82	0.91	0.58	0.86	0.68	13.3 (5.62–33.2)	3.2 × 10^−11^	48	139
**(B) African American Patients Undergoing Multiple Biopsies**										
**Model**	**Cutoff**	**AUC**	**Sensitivity**	**Specificity**	**PPV**	**NPV**	**OR (CI)**	***p*-value**	**nBPH**	**nPCa**
PSA	0.84	0.6	0.91	0.2	0.86	0.28	2.41 (0.606–8.24)	0.16	25	139
Age	0.7	0.82	0.91	0.52	0.91	0.5	10.3 (3.54–30.8)	3.1 × 10^−6^	25	139
FLNA	0.72	0.81	0.96	0.6	0.93	0.71	31.7 (9.33–125)	1.8 × 10^−10^	25	139
PSA + Age	0.69	0.82	0.91	0.48	0.91	0.48	8.76 (3–26.2)	1.7 × 10^−5^	25	139
FLNA + Age	0.7	0.89	0.91	0.6	0.93	0.54	14.1 (4.87–43.6)	7.7 × 10^−8^	25	139
FLNA + Age + Vol	0.72	0.89	0.91	0.68	0.94	0.57	19.9 (6.7–65)	1.2 × 10^−9^	25	139
**(C) African American Patients Comparing BPH vs. PCa Gleason Score 7–10**										
**Model**	**Cutoff**	**AUC**	**Sensitivity**	**Specificity**	**PPV**	**NPV**	**OR (CI)**	***p*-value**	**nBPH**	**nPCa**
PSA	0.55	0.5	0.92	0.17	0.58	0.62	2.18 (0.579–9.15)	0.24	48	60
Age	0.39	0.71	0.92	0.38	0.65	0.78	6.48 (2.06–24.6)	0.00031	48	60
FLNA	0.41	0.71	0.95	0.4	0.66	0.86	12.2 (3.2–69.3)	1.7 × 10^−5^	48	60
PSA + Age	0.35	0.72	0.92	0.38	0.65	0.78	6.48 (2.06–24.6)	0.00031	48	60
FLNA + Age	0.38	0.77	0.92	0.46	0.68	0.81	9.1 (2.94–34.3)	1.1 × 10^−5^	48	60
FLNA + Age + Vol	0.4	0.82	0.92	0.54	0.71	0.84	12.6 (4.1–47.7)	2.7 × 10^−7^	48	60

**Table 4 cancers-16-00712-t004:** Results of logistic model to distinguish patients with PCa from patients with BPH among CA patients for (A) 281 patients with PCa from 191 patients with BPH, (B) 281 patients with PCa from 67 patients with BPH with multiple negative biopsies, (C) 109 patients with Gleason Scores 7–10 PCa from 191 patients with BPH.

(A) Caucasian Patients Comparing BPH vs. PCa										
Model	Cutoff	AUC	Sensitivity	Specificity	PPV	NPV	OR (CI)	*p*-value	nBPH	nPCa
PSA	0.57	0.56	0.9	0.058	0.58	0.28	0.553 (0.242–1.18)	0.13	191	281
Age	0.51	0.58	0.91	0.2	0.63	0.61	2.65 (1.49–4.82)	0.00046	191	281
FLNA	0.47	0.68	0.91	0.36	0.68	0.73	5.64 (3.33–9.79)	1.6 × 10^−12^	191	281
PSA + Age	0.52	0.58	0.9	0.19	0.62	0.57	2.17 (1.24–3.84)	0.0042	191	281
FLNA + Age	0.46	0.68	0.9	0.37	0.68	0.71	5.21 (3.13–8.85)	5 × 10^−12^	191	281
FLNA + Age + Vol	0.49	0.76	0.9	0.46	0.71	0.76	7.68 (4.66–13)	4.4 × 10^−19^	191	281
**(B) Caucasian Patients Undergoing Multiple Biopsies**										
**Model**	**Cutoff**	**AUC**	**Sensitivity**	**Specificity**	**PPV**	**NPV**	**OR (CI)**	***p*-value**	**nBPH**	**nPCa**
PSA	0.77	0.59	0.9	0.13	0.81	0.24	1.4 (0.551–3.26)	0.39	67	281
Age	0.69	0.67	0.91	0.3	0.85	0.45	4.53 (2.19–9.35)	1.7 × 10^−5^	67	281
FLNA	0.7	0.84	0.91	0.64	0.91	0.63	18.1 (9.15–36.9)	1.5 × 10^−20^	67	281
PSA + Age	0.71	0.68	0.9	0.33	0.85	0.44	4.39 (2.19–8.77)	1 × 10^−5^	67	281
FLNA + Age	0.67	0.85	0.9	0.64	0.91	0.61	16 (8.19–32.1)	2 × 10^−19^	67	281
FLNA + Age + Vol	0.69	0.89	0.9	0.7	0.93	0.63	20.9 (10.5–43)	6 × 10^−23^	67	281
**(C) Caucasian Patients Comparing BPH vs. PCa Gleason Score 7–10**										
**Model**	**Cutoff**	**AUC**	**Sensitivity**	**Specificity**	**PPV**	**NPV**	**OR (CI)**	***p*-value**	**nBPH**	**nPCa**
PSA	0.35	0.46	0.91	0.031	0.35	0.38	0.322 (0.0934–1.01)	0.033	191	109
Age	0.33	0.53	0.91	0.13	0.37	0.71	1.49 (0.657–3.63)	0.35	191	109
FLNA	0.21	0.74	0.97	0.36	0.46	0.96	19.4 (6.08–99.3)	1.6 × 10^−12^	191	109
PSA + Age	0.33	0.52	0.91	0.12	0.37	0.7	1.35 (0.59–3.33)	0.57	191	109
FLNA + Age	0.21	0.74	0.91	0.37	0.45	0.88	5.7 (2.74–13.1)	7.3 × 10^−8^	191	109
FLNA + Age + Vol	0.25	0.8	0.91	0.49	0.51	0.9	9.53 (4.61–21.8)	1.8 × 10^−13^	191	109

## Data Availability

The data can be shared up on request.

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
