# Peer review of "Filamin A Is a Prognostic Serum Biomarker for Differentiating Benign Prostatic Hyperplasia from Prostate Cancer in Caucasian and African American Men"

_cancers, 2024, doi:10.3390/cancers16040712_

Round 1

Reviewer 1 Report (Previous Reviewer 1)

Comments and Suggestions for Authors

The authors appropriately addressed the comments of my previous review. Thank you.

Reviewer 2 Report (New Reviewer)

Comments and Suggestions for Authors

This study demonstrates clinical actionability of FLNA alone without requirement of prostate volume measurement as a test with utility in African American and Caucasian men and represents a significant opportunity to decrease the number of unnecessary biopsies in aggressive prostate cancer diagnosis. The finding is novel and the results support their conclusion. Therefore, I am happy to recommend for publication.

This manuscript is a resubmission of an earlier submission. The following is a list of the peer review reports and author responses from that submission.

Round 1

Reviewer 1 Report

Comments and Suggestions for Authors

The authors showed that a cleaved fragment Filamin A (FLNA) protein measured in human serum by IP-MRM mass spectrometry method can differentiate BPH from prostate cancer. Although the conclusion of the manuscript is supported by the statistical analysis, the future clinical utility of FLNA as a biomarker needs more biological studies of FLNA in relation to BPH and prostate cancer. In this regard, the method for FLNA cleaved peptide measurement is not clearly described. For example, human FLNA (longer isoform) protein contains 2647 amino acids. The authors should describe which part of FLNA protein is identified in human serum, and whether any variability of the FLNA peptides is found in the human serum. In addition, the authors should describe the antibody that was used in the immunoprecipitation experiment regarding its specificity of FLNA protein region. 

Comments on the Quality of English Language

NA

Reviewer 2 Report

Comments and Suggestions for Authors

SUMMARY

  • The authors present a manuscript proposing Filamin A as a diagnostic serum biomarker for differentiating BPH from PCA. The topic is important, given the preponderance of BPH in the male population over the age of 50 and the resultant risks of over-treatment of false positives. Unfortunately, the authors have presented Filamin A as a diagnostic serum biomarker for differentiating BPH from PCA before, in their 2021 paper (Scientific Reports), using the EXACT same data from the EXACT same participant samples and with an almost identical conclusion. Whilst reanalyses of existing data have value, whether this paper reaches the threshold of significant new insight is questionable at best.
  • In particular, the previous 2021 paper reported reasonable AUCs for this exact cohort with a test using Age, FLNA, and prostate volume, for example, an AUC of 0.75 for BPH versus PCA in the 2021 paper. This 2023 manuscript reports consistently worse AUCs for the same cohort by removing the originally proposed biomarkers, e.g. the AUC for FLNA alone is worse than the panel including all 3 markers, falling from 0.75 to 0.65. Is this really a novel reanalysis that offers new insight, that removing markers from a proposed panel leads to reduced performance? It would be remarkable and relevant if removing markers improved performance, but that is not the case here.
  • For this study to offer insight, either a NEW cohort of patient data should be investigated as a validation study or a more meaningful reanalysis should be attempted other than just showing that removing biomarkers from the author's own previously published panel leads to worse AUCs. This would surely be true for virtually every biomarker panel ever published!
  •  

ABSTRACT

  • Suggest instead of "To this extent, we recently identified..." to use "To address this issue, in a previous work we identified...". As currently written it is ambiguous whether the 2021 reference covers the identification of the biomarker or the IP-MRM technique.
  • The Abstract contains far too many numbers which makes it markedly less readable. For each test, there is no need to report the full details of AUC, PPV, NPV, OR, and p-value. Suggest reporting only AUC and OR in the Abstract as the full metrics are reported in the manuscript.

INTRODUCTION / REFERENCES

  • The Introduction is inadequate. 2 years have elapsed since the previous paper the authors published, and many manuscripts have been published on prostate cancer and BPH since then. This ties to a wider issue. Only 15 references, most of which are out-of-date (only 2 are more recent than 2018, one of which is the author's own referenced work). A good Introduction sets out the current context for the authors' work. This Introduction does not do that.
  •  

METHODS

  • Basic patient metadata should be presented. I recognize that the authors have provided a reference, but readers should not have to follow references to access basic patient metadata for a manuscript. Suggest that the authors reproduce Table 1 from their 2021 work.
  • I am not sure why the authors refer to parsimonious logistic regression. This is not a scientific or reproducible term. If the authors wish to use a model with fewer features, use why not use LASSO regularization?
  • Also in Methods, it would be helpful to stress that the population has already been extensively analyzed in the authors' previous paper. The Methods should stress what is NEW and NOVEL in this paper, which in the main is the ethnicity split, and the removal of biomarkers from the previously proposed three-marker panel.
  • It is also not clear whether the cohorts were separated into test:train populations for the purposes of logistic regression analysis, and if not, why not. For reproducibility, the language and libraries should also be specified and cited (e.g. Python and scikit learn) with their version numbers.

RESULTS

  • Please consider some visualizations to aid the reader. For example, a plot of AUCs across all the tests, faceted by AA and CA. This provides a more accessible way to compare the various results reported than the tables.

DISCUSSION

  • See previous comments on the Introduction. The authors do very little to compare their findings to existing research. Why have the authors not discussed what role or niche their test would fulfill compared to - for example - MiPS, Stockholm-3, or Proclarix? The authors mention new technologies, but why not SWATH-MS, widely used for cancer proteomic biomarker research? The study also lacks an appropriate discussion of limitations, for example, the lack of healthy controls. How many healthy controls would be classified as PCa using this model? What does this suggest about specificity?
Comments on the Quality of English Language

It is readable but needs improvement.